# EFFICIENT MODEL-BASED DEEP LEARNING VIA NETWORK PRUNING

## ABSTRACT

There is a growing interest in model-based deep learning (MBDL) for solving imaging inverse problems. MBDL networks can be seen as iterative algorithms that estimate the desired image using a physical measurement model and a learned image prior specified using a convolutional neural net (CNNs). The iterative nature of MBDL networks increases the test-time computational complexity, which limits their applicability in certain large-scale applications. We address this issue by presenting *structured pruning algorithm for model-based deep learning (SPADE)* as the first application of structured pruning for MBDL networks. SPADE reduces the computational complexity of CNNs used within MBDL networks by pruning its non-essential weights. We propose three distinct strategies to fine-tune the pruned MBDL networks to minimize the performance loss. Each fine-tuning strategy has a unique benefit that depends on the presence of a pre-trained model and a high-quality ground truth. We validate SPADE on two distinct inverse problems, namely compressed sensing MRI and image super-resolution. Our results highlight that MBDL models pruned by SPADE can achieve substantial speed up in testing time while maintaining competitive performance.

## 1 INTRODUCTION

The recovery of unknown images from noisy measurements is one of the most widely-studied problems in computational imaging. This task is often known as *inverse problems*. Conventional methods solve these problems by formulating optimization problems that consist of a data fidelity term enforcing consistency with the measurements and a regularizer imposing prior knowledge of the unknown images (Hu et al., 2012; Elad & Aharon, 2006; Rudin et al., 1992). The focus in the area has recently shifted to methods based on *deep learning (DL)* (Gilton et al., 2020; Lucas et al., 2018; McCann et al., 2017). A widely-used approach involves training a *convolutional neural network (CNN)* to map the measurements directly to a high-quality reference in an end-to-end fashion (Kang et al., 2017; Chen et al., 2017; Wang et al., 2016).

Model-based deep learning (MBDL) has emerged as an alternative to traditional DL (Ongie et al., 2020; Kamilov et al., 2023; Monga et al., 2021). The key idea behind MBDL is to iteratively update images through operators that integrate the measurement models of the imaging systems and the learned CNNs. Notable examples of MBDL include *plug-and-play (PnP)* (Venkatakrishnan et al., 2013; Sreehari et al., 2016), *regularization by denoising (RED)* (Romano et al., 2017), *deep unfolding (DU)* (Schlemper et al., 2018; Yang et al., 2016; Hammernik et al., 2018) and *deep equilibrium models (DEQ)* (Gilton et al., 2021; Heaton et al., 2021). Despite its superior performance, the iterative nature of MBDL also results in high computational cost during testing, limiting its applicability in large-scale applications. The computational complexity of MBDL arises from both the measurement models and the learned CNNs within the operators. Although several studies in MBDL have reduced the computational demand of the measurement models (Liu et al., 2022; Wu et al., 2020; Sun et al., 2019; Liu et al., 2021; Tang & Davies, 2020), to the best of our knowledge, effort to mitigate the computational cost from the standpoint of the CNN priors remains unexplored.

In this paper, we bridge this gap by proposing a *novel application of **S**tructured **P**runing **A**lgorithm for model-based **DE**ep learning (SPADE)*. SPADE uses the group $\ell_1$-norm criteria to rank the importance of filters in the pre-trained CNN and then progressively eliminates filters from the least to the most important. We propose three distinct learning strategies for fine-tuning the pruned models

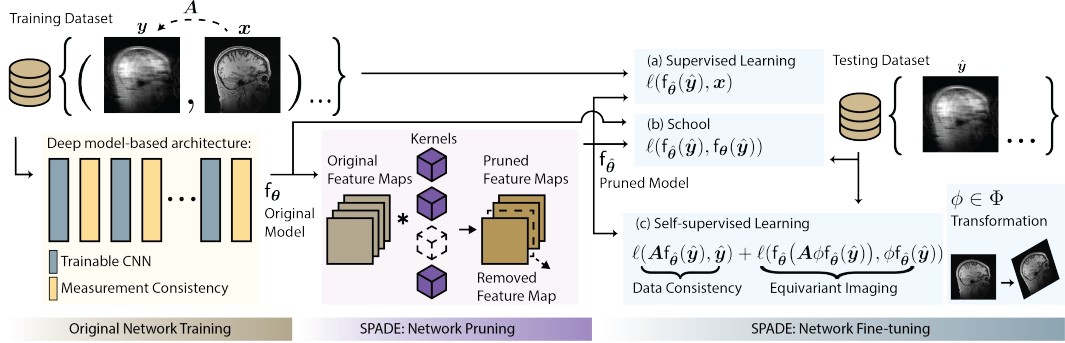

**Figure 1:** An illustration of the pipeline of SPADE. SPADE consists of two components (see Section 3): *(a)* a structured pruning algorithm that physically removes CNN filters based on the group $\ell_1$-norm, and *(b)* fine-tuning algorithms to minimize the performance loss between the pre-trained model and the pruned model. Each fine-tuning strategy has unique applicability depending on the presence of the pre-trained models and high-quality ground truth.

depending on the availability of pre-trained models and ground truth: *(a) supervised* penalizes the discrepancy between pruned model output and corresponding ground truth; *(b) school* enforces consistency between pruned model output and that of the pre-trained model; *(c) self-supervised* relies exclusively on testing dataset by using the losses of data fidelity and equivariant imaging (Chen et al., 2021). We validated SPADE on two imaging problems: compressed sensing MRI (CS-MRI) and image super-resolution. We conducted comprehensive experiments to demonstrate the effectiveness of SPADE.

## 2 BACKGROUND

**Inverse Problems.** Consider imaging inverse problems that aim to recover unknown images $\boldsymbol{x} \in \mathbb{R}^n$ from noisy measurements $\boldsymbol{y} \in \mathbb{R}^m$ characterized by a linear system

$$\boldsymbol{y} = \boldsymbol{A}\boldsymbol{x} + \boldsymbol{e} \,, \tag{1}$$

where $\boldsymbol{A} \in \mathbb{R}^{m \times n}$ represents the measurement model of the imaging system, and $\boldsymbol{e} \in \mathbb{R}^m$ denotes an *additive white Gaussian noise (AWGN)* vector. Due to the noise perturbation and ill-posedness (*i.e.,* $m \ll n$), it is common to solve this problem by formulating an optimization problem

$$\widehat{\boldsymbol{x}} \in \underset{\boldsymbol{x} \in \mathbb{R}^n}{\arg\min} f(\boldsymbol{x}) \text{ with } f(\boldsymbol{x}) = h(\boldsymbol{x}) + g(\boldsymbol{x}) \,, \tag{2}$$

where $h(\boldsymbol{x})$ denotes the data-fidelity term that quantifies consistency with the measurements $\boldsymbol{y}$, and $g(\boldsymbol{x})$ is the regularizer that imposes a prior on $\boldsymbol{x}$. For example, a widely-used data-fidelity term and regularizer in imaging are least-square $h(\boldsymbol{x}) = \frac{1}{2} \|\boldsymbol{A}\boldsymbol{x} - \boldsymbol{y}\|_2^2$ and total variation $g(\boldsymbol{x}) = \tau \|\boldsymbol{D}\boldsymbol{x}\|_1$ where $\boldsymbol{D}$ is the gradient operator, and $\tau$ is the trade-off parameter.

**DL and MBDL.** There is a growing interest in DL for solving inverse problems due to its excellent performance (see reviews in (Gilton et al., 2020; Lucas et al., 2018; McCann et al., 2017)). A widely-used approach in this context trains a CNN to directly learn a regularized inversion that maps the measurements to high-quality reference (Kang et al., 2017; Chen et al., 2017; Wang et al., 2016; Zhu et al., 2018). MBDL has emerged as powerful DL framework for inverse problems by integrating the measurement models and learned CNNs (see also reviews in (Ongie et al., 2020; Kamilov et al., 2023; Monga et al., 2021)). Notable examples of MBDL include *plug-and-play (PnP)* (Venkatakrishnan et al., 2013; Sreehari et al., 2016), *regularization by denoising (RED)* (Romano et al., 2017), *deep unfolding (DU)* (Schlemper et al., 2018; Yang et al., 2016; Hammernik et al., 2018) and *deep equilibrium models (DEQ)* (Gilton et al., 2021; Heaton et al., 2021). PnP/RED refers to a family of algorithms that consider a CNN denoiser $\mathsf{D}_{\boldsymbol{\theta}}$ parameterized by $\boldsymbol{\theta}$ as the imaging prior and then use $\mathsf{D}_{\boldsymbol{\theta}}$ in fixed-point iterations of some high-dimensional operators $\mathsf{T}_{\boldsymbol{\theta}}$. For example, the fixed-point iteration of PnP proximal gradient method is formulated as

$$\boldsymbol{x}^{k+1} = \mathsf{T}_{\boldsymbol{\theta}}(\boldsymbol{x}^k) = \mathsf{D}_{\boldsymbol{\theta}}(\boldsymbol{x}^k - \gamma \cdot \nabla h(\boldsymbol{x}^k)) \,, \tag{3}$$

where $\gamma > 0$ denotes the step size, and $k = 0, ..., K$. DU denotes a special end-to-end network architecture obtained by interpreting a *finite* iteration of PnP/RED as different layers of the network. DEQ (Bai et al., 2019; 2020) is a recent approach that allows training *infinite-depth*, *weight-tied* networks by analytically backpropagating through the fixed points using implicit differentiation. Training DEQ for inverse problems is equivalent to training a weight-tied DU with infinite iterations. To be specific, the forward pass of DEQ estimates a fixed-point $\bar{\boldsymbol{x}}$ of the operator $\bar{\boldsymbol{x}} = \mathsf{T}_{\boldsymbol{\theta}}(\bar{\boldsymbol{x}})$. The backward pass of DEQ updates $\mathsf{D}_{\boldsymbol{\theta}}$ by computing an implicit gradient of the training loss $\ell$

$$\nabla \ell(\boldsymbol{\theta}) = \left(\nabla_{\boldsymbol{\theta}} \mathsf{T}(\bar{\boldsymbol{x}})\right)^{\mathsf{T}} \left(\mathsf{I} - \nabla_{\boldsymbol{x}} \mathsf{T}(\bar{\boldsymbol{x}})\right)^{-\mathsf{T}} \nabla \ell(\bar{\boldsymbol{x}}). \tag{4}$$

MBDL has exhibited excellent performances in many imaging problems, such as MRI (Gan et al., 2020; Liu et al., 2020; Sriram et al., 2020; Cui et al., 2023; Hammernik et al., 2021; Schlemper et al., 2018; Hu et al., 2022), CT (Adler & Oktem, 2018; Wu et al., 2021; Liu et al., 2021; 2022; He et al., 2019a), and image restoration (Zhang et al., 2021; 2020; Gilton et al., 2021). However, its iterative nature results in a high computational cost due to multiple CNN applications, limiting its use in large-scale or computation-constrained applications. While numerous studies have tackled the computational demand associated with the measurement models (Liu et al., 2022; Wu et al., 2020; Sun et al., 2019; Liu et al., 2021; Tang & Davies, 2020), to the best of our knowledge, the reduction of the computational cost from the perspective of the CNN priors has not yet been explored.

**Network Pruning.** Network pruning denotes the process of eliminating weights, filters, or channels of a pre-trained network to obtain lightweight models (see also recent reviews in (Ghimire et al., 2022; He & Xiao, 2023; Cheng et al., 2023; Hoefler et al., 2021)). Pruning methods can be divided into unstructured pruning and structured pruning. Unstructured pruning *virtually* masks out unimportant individual weights throughout the network. Since the unimportant weights are not removed physically, specialized software or hardware is required for computational acceleration in structural pruning (Zhang et al., 2016; Parashar et al., 2017; Zhou et al., 2018a; Chen et al., 2019). Structured pruning, on the other hand, *physically* removes entire filters, channels, or layers, leading to faster inference without the need of any specialized hardware or software. Many approaches have been proposed to identify the importance of the network filters prior to removing any of them, including those are based on *(a)* certain criteria of the filters (He et al., 2018; Lin et al., 2020; Hu et al., 2016; Li et al., 2017), such as $\ell_1$-norm (Li et al., 2017), *(b)* minimizing the reconstruction errors (Luo et al., 2017; Yu et al., 2018), or *(c)* finding the replaceable filters with similarity measurements (He et al., 2019b; Zhou et al., 2018b).

**Fine-tuning Pruned Networks.** Following the pruning of the network, it is common to fine-tune the pruned model to minimize performance degradation (see also Section 2.4.6 in (Hoefler et al., 2021)). A widely-used strategy is to use the same amount of training data for the pre-trained models to retrain pruned models (Han et al., 2015; Hu et al., 2016; Li et al., 2017; Luo et al., 2017; He et al., 2017; Yu et al., 2018; Zhou et al., 2018b; Lin et al., 2020; Lee & Song, 2022). In imaging inverse problems, high-quality ground truth is commonly considered as the learning target for CNNs. However, ground truth data is not always available in practice, which limits the applicability of this fine-tuning approach in MBDL models. Other fine-tuning approaches include re-initialization of the pruned model's weight based on lottery-ticket-hypothesis (Frankle & Carbin, 2018), and *knowledge distillation (KD)* that configures the pre-trained and pruned networks as a teacher-student pair (Hinton et al., 2015; Dong & Yang, 2019; Mirzadeh et al., 2020; Lee & Song, 2022). For example, KD in (Dong & Yang, 2019) proposes an auxiliary loss function to match the prediction of a pruned network and soft targets unpruned network.

**Self-supervised Deep Image Reconstruction.** There is a growing interest in developing DL methods that reduce the dependence on the ground truth data (see recent reviews in (Akçakaya et al., 2022; Tachella et al., 2023; Zeng et al., 2021)). Some widely-used strategies include *Noise2Noise (N2N)* (Lehtinen et al., 2018; Gan et al., 2022), *Noise2Void (N2V)* (Krull et al., 2019), *deep image prior (DIP)* (Ulyanov et al., 2018), *compressive sensing using generative model (GSGM)* (Bora et al., 2018; Gupta et al., 2021), and *equivariant imaging (EI)* (Chen et al., 2021). In particular, EI assumes the set of ground truth is invariant to a certain group of transformations $\Phi$. The training loss of EI can then be formulated as

$$\ell_{\mathsf{EI}}(\boldsymbol{\theta}) = \ell\left(\mathsf{f}_{\boldsymbol{\theta}}(\boldsymbol{A}\widehat{\boldsymbol{x}}_\phi), \widehat{\boldsymbol{x}}_\phi\right) \text{ with } \widehat{\boldsymbol{x}}_\phi = \phi \, \mathsf{f}_{\boldsymbol{\theta}}(\boldsymbol{y}) \,, \tag{5}$$

where $\mathsf{f}_{\boldsymbol{\theta}}$ denotes the DL model, and $\phi \in \Phi$ is an instance of the transformation. The effectiveness of EI has been validated in a variety of imaging (Chen et al., 2021; 2022), such as sparse-view CT

and image inpainting. EI can also be integrated with another training loss, such as data-fidelity loss and adversarial loss (Mao et al., 2017). As can be seen in the next section, we exploit EI to fine-tune our pruned model using exclusively the testing dataset.

**Our Contributions**: *(1)* We propose the first network pruning algorithm specifically designed for MBDL models, aiming to reduce computational complexity at testing time. While the technique of network pruning has been extensively explored across a variety of tasks in computer vision, its potential has remained unexplored in the realm of imaging inverse problems; *(2)* We develop three distinct fine-tuning methods to minimize the performance gap between pre-trained and pruned models. Each of these methods, to be detailed in the following section, is intuitive and holds practical applicability for inverse problems; *(3)* We have conducted comprehensive experiments across various imaging problems, diverse MBDL methods, and different pruning ratios. Such extensive numerical validations represent a novel contribution, as they have not been performed in prior works.

## 3 STRUCTURED PRUNING ALGORITHM FOR MBDL

As illustrated in Figure 1, SPADE consists of a filter-pruning method and several fine-tuning methods. For the filter pruning method, we adopt DepGraph (Fang et al., 2023) to identify layer dependencies and form *layer groups* across the network. Let $\mathsf{f}_{\boldsymbol{\theta}}$ denote the original unpruned model with $N$ layers. Let also $\mathsf{f}_{\boldsymbol{\theta},j}^-$ and $\mathsf{f}_{\boldsymbol{\theta},j}^+$ denote the input and the output of the $j$th layer $\mathsf{f}_{\boldsymbol{\theta},j}$, respectively. Consider two types of dependencies between $\mathsf{f}_{\boldsymbol{\theta},j}^-$ and $\mathsf{f}_{\boldsymbol{\theta},i}^+$ for all $i, j = 1, \ldots, N$: *(a)* inter-layer dependency for $i \neq j$ indicates that $\mathsf{f}_{\boldsymbol{\theta},j}^-$ and $\mathsf{f}_{\boldsymbol{\theta},i}^+$ are topologically connected and correspond to the same intermediate features of the network, and *(b)* intra-layer dependency for $i = j$ exists if and only if the mapping from $\mathsf{f}_{\boldsymbol{\theta},j}^-$ to $\mathsf{f}_{\boldsymbol{\theta},i}^+$ can be expressed as a diagonal matrix. Another conceptual interpretation of intra-layer dependency is that $\mathsf{f}_{\boldsymbol{\theta},j}^+$ and $\mathsf{f}_{\boldsymbol{\theta},i}^-$ *share the same pruning scheme* (Fang et al., 2023). For example, consider a convolutional layer (Conv). Consider a filter $\boldsymbol{K} \in \mathbb{R}^{K_{\mathsf{in}} \times K_{\mathsf{out}} \times H \times W}$ in a Conv, where $K_{\mathsf{in}}$ denotes the number of input channels, $K_{\mathsf{out}}$ is the number of output channels, and $H \times W$ represents the kernel size. Pruning the input of a Conv necessitates pruning the kernel along the $K_{\mathsf{in}}$ dimension. Conversely, pruning the output demands alterations along the $K_{\mathsf{out}}$ dimension. The difference in pruning schemes for the input and the output of Conv indicates the absence of an intra-layer dependency. DepGraph examines all inputs and output pairs to compute their dependencies (see also Algorithm 1 in (Fang et al., 2023)).

The layer groups across the network can then be constructed based on the identified dependencies (also refer to Algorithm 2 in (Fang et al., 2023)). Each group must adhere to the following conditions: *(a)* the group can be represented as a connected graph, where the nodes are the layers within the group, and the edges denote the topological connections between the input and output of two layers (*i.e.,* inter-layer dependency); *(b)* the hidden layers (*i.e.,* non-edge layers) within the group must exhibit intra-layer dependencies. Note that the inter-layer and intra-layer dependencies ensure that the prunable dimensions, such as the number of feature map channels in CNN, are identical across different layers in the same group. For example, consider a sample network of $\{\mathsf{Conv}_1 \rightarrow \mathsf{BN}_1 \rightarrow \mathsf{Conv}_2 \rightarrow \mathsf{BN}_2\}$, where $\rightarrow$ denotes topological connection, and BN is the batch normalization layer. The resulting layer groups include $\{\mathsf{Conv}_1, \mathsf{BN}_1, \mathsf{Conv}_2\}$ and $\{\mathsf{Conv}_2, \mathsf{BN}_2\}$. The prunable dimension of the first and the second group match the output channel of $\mathsf{Conv}_1$ (or the input channel of $\mathsf{Conv}_2$) and the output channel of $\mathsf{Conv}_2$, respectively.

We use the group $\ell_1$-norm to evaluate the importance of parameters in each layer group. To be specific, consider a layer group with $M$ layers with $i = 1, ..., M$ denoting the $i$th layer within the group. Let also $\boldsymbol{w}_k^i$ be the filters of the $k$th prunable dimension in the $i$th layer for $k = 1, ..., K$. The group $\ell_1$ norm vector of a layer group is formulated as $\boldsymbol{\alpha} = [\alpha_1, \cdots, \alpha_k]$ where

$$\alpha_k = \frac{1}{M} \sum_{i=1}^{M} \left\| \boldsymbol{w}_k^i \right\|_1, \quad k = 1, \ldots, K . \tag{6}$$

For each layer group, we compute the group $\ell_1$ norm and subsequently select a subset of the smallest $\alpha_k$ according to a pre-defined pruning ratio. Parameters associated with this subset are then pruned.

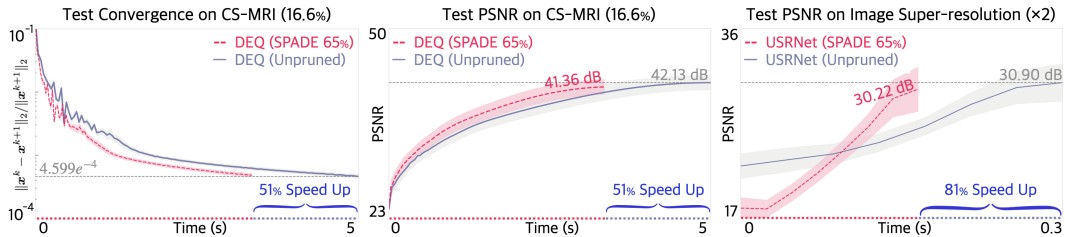

**Figure 2:** Illustration of testing time evolution between MBDL models and the pruned variants using SPADE. *Left:* evolution of the distance between two consecutive images; *Middle and Right:* evolution of the testing PSNR values compared to the ground truth. We used the *supervised* and *school* strategies to fine-tune DEQ and USRNet, respectively. Note how models pruned by SPADE can significantly reduce the testing time while maintaining competitive performance.

Let $f_{\hat{\theta}}$ be the pruned model. SPADE considers three distinct fine-tuning algorithms to minimize the performance loss between $f_{\hat{\theta}}$ and $f_{\theta}$. Each fine-tuning method has unique applicability depending on the presence of the unpruned model and high-quality ground truth.

*(a)* **supervised (SV):** SV considers a training set consisting pairs of measurements and ground truth $\{y_i, x_i\}_{i=1}^N$, where $N$ denotes the total number of training samples. The loss function of SV minimizes the difference between the reconstruction of the pruned model and ground truth

$$\ell_{\text{sv}}(\hat{\theta}) = \sum_{i=1}^N \ell(f_{\hat{\theta}}(y_i),\ x_i) \ . \tag{7}$$

This training strategy is also the most common scheme for training CNNs for imaging inverse problems starting from scratch. Despite its effectiveness (see Sec. 4.1), SV relies on a collection of high-quality ground truth, which might not be always available in practice.

*(b)* **school (SC):** SC considers a testing set of unseen measurements $\{y_j\}_{j=1}^M$ where $M$ denotes the total number of testing samples. The loss function of SC penalizes the discrepancy between the outputs of pruned model and those of unpruned model

$$\ell_{\text{sc}}(\hat{\theta}) = \sum_{j=1}^M \ell(f_{\hat{\theta}}(y_j),\ f_{\theta}(y_j)) \ . \tag{8}$$

SC can be seen as a new branch of KD in imaging, where we transfer the "knowledge" from the unpruned model (*teacher*) to the pruned model (*student*). The reason why it is called school follows. Unlike SV, SC only requires access of the pre-trained model.

*(c)* **self-supervised (SS):** Compared to SC, SS considers a particular case where the pre-trained models are unavailable for some reason (e.g., privacy). The key idea behind SS is to fine-tune $f_{\hat{\theta}}$ using exclusively the testing set $\{y_j\}_{j=1}^M$. Let $\Phi$ denote a certain group of transformations. The loss function of SS can be formulated as

$$\ell_{\text{ss}}(\hat{\theta}) = \sum_j^M \underbrace{\ell(A_j f_{\hat{\theta}}(y_j),\ y_j)}_{\text{Data Fidelity}} + \underbrace{\ell\big(f_{\hat{\theta}}(A_j \widehat{x}_{\phi,j}),\ \widehat{x}_{\phi,j}\big)}_{\text{Equivariant Imaging}} \ . \tag{9}$$

where $\widehat{x}_{\phi,j} = \phi_j\, f_{\hat{\theta}}(y_j)$, and $\phi_j \sim \Phi$ is a transformation sampled randomly from $\Phi$.

## 4 NUMERICAL RESULTS

The goal of our experiments was to validate the effectiveness of SPADE on different imaging problems, different MBDL networks, and different pruning ratios. We tested SPADE on two imaging problems: CS-MRI and image super-resolution. We used *peak signal-to-noise ratio (PSNR)* and *structural similarity index (SSIM)* for quantitative evaluation. We pre-defined pruning ratios of 5%, 10%, 20%, and 40%, resulting in actual pruning ratios of 10%, 20%, 35%, and 65% since the layers

**Table 1:** Quantitative evaluation of MBDL models pruned by SPADE with different fine-tuning strategies in CS-MRI at the sampling rate of 16.6 %.

| Network | Pruning Ratio | PSNR (dB) | | | SSIM (%) | | | Time (ms) | Speed Up | # Params |
|---|---|---|---|---|---|---|---|---|---|---|
| | | Supervised | School | Self-supervised | Supervised | School | Self-supervised | | | |
| DEQ | 0 % | | 42.13 | | | 98.7 | | 4954.30 | ×1.00 | 999,428 |
| | 10 % | 42.02 | 41.85 | 40.37 | 98.6 | 98.6 | 98.2 | 4812.57 | ×1.03 | 878,644 |
| | 20 % | 42.02 | 41.69 | 38.79 | 98.7 | 98.6 | 97.7 | 4708.00 | ×1.06 | 793,159 |
| | 35 % | 41.94 | 41.32 | 36.14 | 98.6 | 98.5 | 96.5 | 4233.70 | ×1.17 | 635,311 |
| | 65 % | 41.36 | 39.99 | 34.54 | 98.5 | 98.2 | 95.7 | 3274.58 | ×1.51 | 353,328 |
| VarNet | 0 % | | 39.25 | | | 97.7 | | 161.6 | ×1.00 | 19,634,712 |
| | 10 % | 39.16 | 38.86 | 39.11 | 97.7 | 97.6 | 97.4 | 161.4 | ×1.00 | 17,567,216 |
| | 20 % | 39.06 | 38.71 | 38.59 | 97.7 | 97.5 | 97.3 | 151.8 | ×1.06 | 15,772,920 |
| | 35 % | 38.85 | 38.54 | 37.46 | 97.6 | 97.4 | 96.9 | 144.7 | ×1.12 | 12,477,088 |
| | 65 % | 38.12 | 37.36 | 34.17 | 97.3 | 96.7 | 95.3 | 122.6 | ×1.32 | 6,988,752 |
| E2EVar | 0 % | | 44.24 | | | 99.2 | | 210.7 | ×1.00 | 20,119,610 |
| | 10 % | 44.18 | 43.76 | 40.12 | 99.2 | 99.1 | 97.7 | 209.3 | ×1.01 | 18,052,114 |
| | 20 % | 44.17 | 43.24 | 39.91 | 99.2 | 99.0 | 97.6 | 201.1 | ×1.05 | 16,257,818 |
| | 35 % | 43.71 | 42.66 | 39.28 | 99.1 | 98.9 | 97.3 | 192.8 | ×1.09 | 12,961,986 |
| | 65 % | 42.62 | 41.58 | 37.82 | 98.8 | 98.5 | 96.9 | 170.4 | ×1.24 | 7,473,650 |

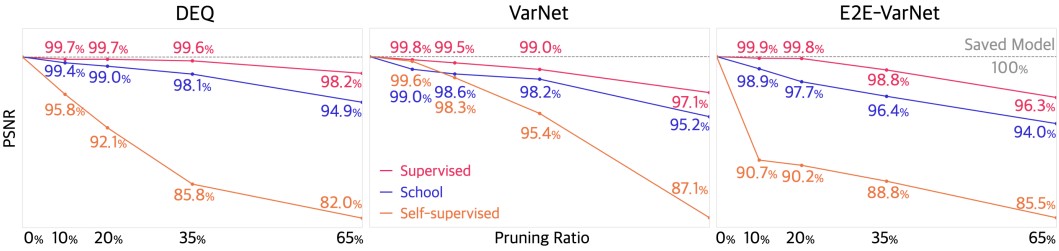

**Figure 3:** Degradation PSNR percentage of pruned MBDL models compared to the unpruned model in different pruning ratios and different fine-tuning strategies. These results correspond to experiments of CS-MRI at the sampling rate of 16.6 %. Note how *supervised* fine-tuning method can reduce 65% parameters while maintaining less than 4% PSNR degradation.

dependent on the pruned layers are also removed. (see Sec. 3). All pre-trained models were trained in a supervised learning manner to ensure their optimal performance. We implemented Φ in (9) as a set of rotations. We pre-trained and fine-tuned the MBDL models by using the Adam optimizer with the learning rate being $10^{-5}$. We conducted all experiments on a machine equipped with an AMD EPYC 7443P 24-Core Processor and 4 NVIDIA RTX A6000 GPUs.

## 4.1 COMPRESSED SENSING MRI

The measurement model of CS-MRI consists of a set of complex measurement operators depending on a set of receiver coils $\{S_i\}$. For each coil, we have $A_i = PFS_i$, where $F$ is the Fourier transform, $P$ denotes the diagonal sampling matrix, and $S_i$ is the diagonal matrix of sensitivity maps. We used T2-weighted MR brain acquisitions of 165 subjects obtained from the validation set of the fastMRI dataset (Knoll et al., 2020) as the fully sampled measurement for simulating measurements. We obtained reference coil sensitivity maps from the fully sampled measurements using ESPIRiT (Uecker et al., 2014). These 165 subjects were split into 145, 10, and 10 for training, validation, and testing, respectively. We followed (Knoll et al., 2020) to retrospectively undersample the fully sampled data using 1D Cartesian equispaced sampling masks with 10% auto-calibration signal (ACS) lines. We conducted our experiments for sampling rate of 16.7% and 12.5%.

We tested SPADE on DEQ and two DU models: VarNet (Hammernik et al., 2018), and E2E-VarNet (Sriram et al., 2020). We implemented DEQ with forward iteration as in (3) and EDSR (Lim et al., 2017) as the CNN architecture. We ran the forward-pass of DEQ with a maximum number of iterations of 100 and the stopping criterion of the relative norm difference between iterations being

**Table 2:** Quantitative evaluation of MBDL models pruned by SPADE with different fine-tuning strategies in CS-MRI at the sampling rate of 12.5 %.

| Network | Pruning Ratio | PSNR (dB) | | | SSIM (%) | | | Time (ms) | Speed Up | # Params |
|---------|---------------|-----------|--------|----------------|----------|--------|----------------|-----------|----------|
| | | Supervised | School | Self-supervised | Supervised | School | Self-supervised | | | |
| DEQ | 0 % | | 38.07 | | | 96.9 | | 4954.30 | ×1.00 | 999,429 |
| | 10 % | 38.06 | 37.87 | 36.13 | 96.9 | 96.9 | 96.2 | 4812.57 | ×1.03 | 878,644 |
| | 20 % | 38.07 | 37.77 | 34.94 | 96.9 | 96.8 | 95.6 | 4708.00 | ×1.06 | 793,159 |
| | 35 % | 37.68 | 37.42 | 32.54 | 96.8 | 96.7 | 93.9 | 4233.70 | ×1.17 | 635,311 |
| | 65 % | 37.13 | 36.33 | 30.26 | 96.5 | 96.2 | 90.4 | 3274.58 | ×1.51 | 353,328 |
| VarNet | 0 % | | 36.17 | | | 96.3 | | 161.6 | ×1.00 | 19,634,712 |
| | 10 % | 36.04 | 36.04 | 35.80 | 96.3 | 96.2 | 95.9 | 161.4 | ×1.00 | 17,567,216 |
| | 20 % | 36.06 | 35.89 | 35.36 | 96.3 | 96.2 | 95.8 | 151.8 | ×1.06 | 15,772,920 |
| | 35 % | 35.80 | 35.55 | 34.69 | 96.1 | 95.9 | 95.2 | 144.7 | ×1.12 | 12,477,088 |
| | 65 % | 35.30 | 34.07 | 31.13 | 95.7 | 94.9 | 92.6 | 122.6 | ×1.32 | 6,988,752 |
| E2EVar | 0 % | | 40.41 | | | 98.0 | | 210.7 | ×1.00 | 20,119,610 |
| | 10 % | 40.31 | 40.00 | 37.31 | 98.0 | 97.9 | 96.4 | 209.3 | ×1.01 | 18,052,114 |
| | 20 % | 40.23 | 39.72 | 37.30 | 98.0 | 97.9 | 96.2 | 201.1 | ×1.05 | 16,257,818 |
| | 35 % | 40.14 | 39.30 | 35.74 | 97.9 | 97.6 | 95.6 | 192.8 | ×1.09 | 12,961,986 |
| | 65 % | 38.93 | 38.43 | 34.62 | 97.4 | 97.1 | 95.0 | 170.4 | ×1.24 | 7,473,650 |

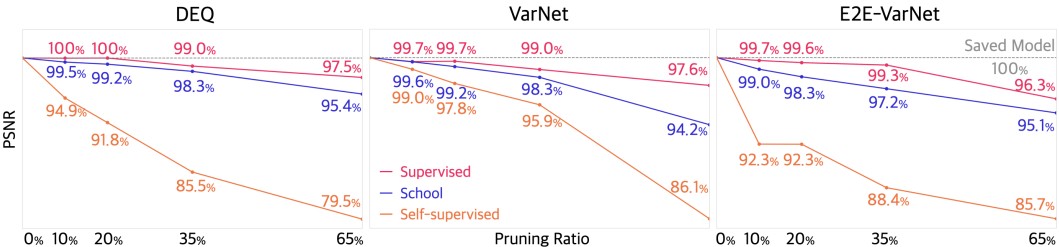

**Figure 4:** Degradation PSNR percentage of pruned MBDL models compared to the unpruned model in different pruning ratios and different fine-tuning strategies. These results correspond to experiments of CS-MRI at the sampling rate of 12.5 %. Note how *supervised* fine-tuning method can reduce 65% parameters while maintaining less than 4% PSNR degradation.

less than $10^{-4}$. The implementations of VarNet and E2E-VarNet were from their official repository[1]. The difference between E2E-VarNet and VarNet is that E2E-VarNet has an additional coil sensitivity estimator compared to VarNet. Noted that SPADE did not prune this estimator for E2E-VarNet. We also used the estimated coil sensitivity map for *self-supervised* fine-tuning. The training of unpruned models for E2E-VarNet and VarNet took around 7 days, and that for DEQ was 10 days.

Table 1 shows quantitative evaluations of MBDL models pruned by SPADE in MRI at a sampling rate of 16.6%. Figure 3 illustrates the PSNR degradation percentage of the pruned model compared to the unpruned network at the same sampling rate. Figure 3 demonstrates that the *supervised* fine-tuning strategy is highly effective, resulting only up to 4% PSNR degradation to achieve approximately 65% fewer parameters and up to ×1.51 speed up. Both Figure 3 and Table 1 also indicate that one can achieve a ×1.06 speed up in testing time with virtually no cost in PSNR degradation by removing 20% of parameters. Moreover, Figure 3 highlights that the *school* fine-tuning method can achieve competitive performance against *supervised* fine-tuning across different pruning ratios. Table 1 further shows that DEQ can achieve higher speed up than DU models under the same pruning ratio, which we attribute to its large number of forward iterations. Table 2 and Figure 4 present similar evaluations as in Table 1 and Figure 3, but at a sampling rate of 12.5%, with consistent observations.

Table 3 presents a quantitative evaluation of pruned MBDL models and equivalently parameterized models trained from scratch on fine-tuning losses of *supervised* and *self-supervised*. Note that the re-trained models use the same loss as the fine-tuned pruned model. The results in Table 3 demonstrate

[1] https://github.com/facebookresearch/fastMRI

**Table 3:** Quantitative evaluation of pruned MBDL models and equivalently parameterized models trained from scratch, shown in SPADE and *Random* columns, respectively. Noted that the retrained models use the same loss when fine-tuning the pruned model. The "Δ" column denotes the quantitative difference between "SPADE" and "*Random*". **(a)**: *self-supervised* fine-tuning strategy. **(b)**: *supervised* fine-tuning strategy. This table demonstrates improved performance by pruning MBDL models over re-training equivalently parameterized models. Note how SPADE with *self-supervised* fine-tuning method can outperform re-trained model at least $8$dB in PSNR.

**(a)**

| Network | Pruning Ratio | PSNR (dB) | | | SSIM (%) | | |
|---|---|---|---|---|---|---|---|
| | | SPADE | *Random* | Δ | SPADE | *Random* | Δ |
| | 0% | 44.24 | | | 99.1 | | |
| E2E-VarNet | 10 % | 40.12 | 29.94 | 10.18 | 97.7 | 90.3 | 7.4 |
| | 20 % | 39.91 | 28.77 | 11.14 | 97.5 | 88.6 | 8.9 |
| | 35 % | 39.28 | 26.74 | 12.54 | 97.3 | 85.7 | 11.6 |
| | 65 % | 37.82 | 29.28 | 8.54 | 96.9 | 90.2 | 6.7 |
| | 0% | 39.25 | | | 97.7 | | |
| VarNet | 10 % | 39.10 | 18.96 | 20.14 | 97.4 | 54.2 | 43.2 |
| | 20 % | 38.59 | 19.11 | 19.48 | 97.3 | 55.0 | 42.3 |
| | 35 % | 37.46 | 17.13 | 20.33 | 96.9 | 51.1 | 45.8 |
| | 65 % | 34.17 | 17.54 | 16.63 | 95.3 | 51.7 | 43.6 |

**(b)**

| Network | Pruning Ratio | PSNR (dB) | | | SSIM (%) | | |
|---|---|---|---|---|---|---|---|
| | | SPADE | *Random* | Δ | SPADE | *Random* | Δ |
| | 0% | 44.24 | | | 99.1 | | |
| E2E-VarNet | 10 % | 44.18 | 42.40 | 1.78 | 99.1 | 98.7 | 0.4 |
| | 20 % | 44.17 | 42.23 | 1.94 | 99.1 | 98.7 | 0.4 |
| | 35 % | 43.71 | 41.69 | 2.02 | 99.1 | 98.4 | 0.7 |
| | 65 % | 42.62 | 41.50 | 1.12 | 98.8 | 98.4 | 0.4 |
| | 0% | 39.25 | | | 97.7 | | |
| VarNet | 10 % | 39.15 | 38.09 | 1.06 | 97.7 | 97.3 | 0.4 |
| | 20 % | 39.06 | 38.11 | 0.95 | 97.7 | 97.2 | 0.5 |
| | 35 % | 38.85 | 38.03 | 0.82 | 97.6 | 97.1 | 0.5 |
| | 65 % | 38.12 | 35.67 | 2.45 | 97.3 | 95.9 | 1.4 |

that pruned models can outperform equivalently parameterized models retrained using the same fine-tuning loss. Note how pruned models with a *self-supervised* fine-tuning strategy can achieve at least an 8 dB improvement in PSNR compared to the retrained networks.

## 4.2 IMAGE SUPER-RESOLUTION

We consider the measurement model of form $A = SH$, where $H \in \mathbb{R}^{n \times n}$ is the blurring matrix, and $S \in \mathbb{R}^{m \times n}$ denotes the standard $d$-fold down-sampling operator with $d^2 = n/m$. We evaluated SPADE on CSDB68 dataset. We conducted our experiments for down-sampling factors of 2 and 3. We followed (Zhang et al., 2020) to experiment with 8 different Gaussian blur kernels and four motion kernels. We tested SPADE on a DU model, USRNet (Zhang et al., 2020), in the *school* fine-tuning method to simulate the circumstance where only pre-trained network is accessible. We used the pre-trained model provided by the official repository[2].

Table 4 shows quantitative evaluation of USRNet pruned by SPADE with *school* fine-tuning strategy in image super-resolution at the scale of $\times 2$ and $\times 3$. This table highlights that the *school* fine-tuning strategy can achieve $\times 1.81$ speed up while maintaining less than $1\%$ degradation in both PSNR and SSIM values. Figure 5 shows visual results of USRNet and its pruned variants in image super-resolution at the scale of $\times 3$. Figure 5 demonstrates that the pruned models can achieve qualitatively competitive performance compared to the unpruned network.

## 5 CONCLUSION

This work proposes SPADE, the first application to reduce the test-time computational complexity of model-based deep learning *through neural network pruning*. SPADE employs group $\ell_1$-norm to identify the significance of CNN weights, pruning them in ascending order of importance. We propose three distinct fine-tuning strategies to minimize the performance deviation between pruned and pre-trained models. Each of these fine-tuning methods possesses unique applications, contingent on the availability of high-quality ground truth and a pre-trained model: *(a) supervised* strategy minimizes the discrepancy between the output of the pruned model and the corresponding ground truth; *(b) school* ensures consistency between the outputs of the pruned and the pre-trained models; *(c) self-supervised* exclusively relies on the testing dataset, leveraging data fidelity and equivariant imaging losses.

---

[2]https://github.com/cszn/USRNet

**Table 4:** Quantitative evaluation of USRNet pruned by SPADE with *school* fine-tuning strategy in image super-resolution at the scale of ×2 and ×3. The "*Degrad. %*" columns denote degradation percentage of PSNR values of pruned models compared to that of pruned models. Note how *school* fine-tuning method can gain 1.81× speed up with less than 1% performance degradation.

| Scale | Pruning Ratio | PSNR (dB) | | SSIM (%) | | Time (ms) | Speed Up | # Params |
|---|---|---|---|---|---|---|---|---|
| | | *School* | *Degrad. %* | *School* | *Degrad. %* | | | |
| | 0 % | 29.96 | 100.0 % | 86.5 | 100.0 % | 272.2 | ×1.00 | 17,016,016 |
| ×2 | 10 % | 29.81 | 99.5 % | 86.2 | 99.6 % | 262.8 | ×1.04 | 15,314,620 |
| | 20 % | 29.81 | 99.5 % | 86.1 | 99.5 % | 248.9 | ×1.09 | 13,730,964 |
| | 35 % | 29.80 | 99.5 % | 86.1 | 99.5 % | 241.8 | ×1.13 | 10,837,246 |
| | 65 % | 29.70 | 99.1 % | 86.0 | 99.4 % | 150.3 | ×1.81 | 6,094,762 |
| | 0 % | 27.56 | 100.0 % | 79.0 | 100.0 % | 272.2 | ×1.00 | 17,016,016 |
| ×3 | 10 % | 27.43 | 99.5 % | 78.5 | 99.3 % | 262.8 | ×1.04 | 15,314,620 |
| | 20 % | 27.43 | 99.5 % | 78.5 | 99.3 % | 248.9 | ×1.09 | 13,730,964 |
| | 35 % | 27.42 | 99.5 % | 78.5 | 99.3 % | 241.8 | ×1.13 | 10,837,246 |
| | 65 % | 27.32 | 99.1 % | 78.2 | 98.9 % | 150.3 | ×1.81 | 6,094,762 |

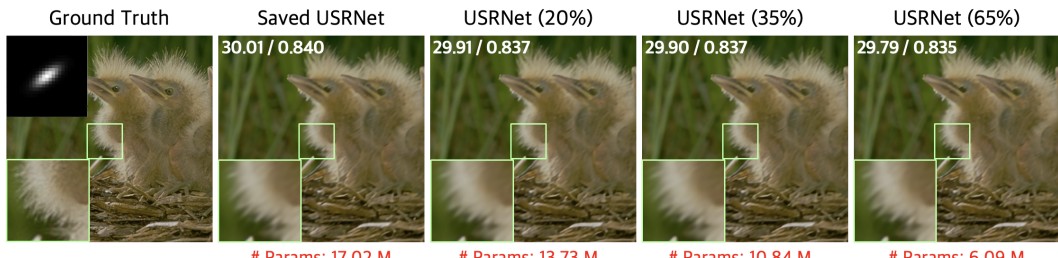

**Figure 5:** Visual results of USRNet and its pruned variants with the *school* fine-tuning strategy at scale of ×3. Note how USRNet has been pruned more than 65% parameters but kept similar visual quality compared to the unpruned model.

We evaluated the efficacy of SPADE through applications in compressed sensing MRI and image super-resolution, employing several MBDL models. The experimental results in MRI demonstrate that: the *(a) supervised* strategy can realize up to ×1.51 speed up in testing time by eliminating 65% of parameters, with less than 4% degradation in testing PSNR values; it can also attain ×1.06 speed up with negligible performance cost; *(b) school* can achieve competitive performance against *supervised*, with less than 3% PSNR degradation across different pruning ratios and MBDL models; *(c) self-supervised* can outperform equivalently parameterized models trained from scratch using the same loss function. The results in image super-resolution further corroborate the effectiveness of SPADE on the *school* fine-tuning method. Future directions for this research include testing SPADE with alternative approaches to ranking the importance of CNN weights, and exploring different losses to enhance the *self-supervised* fine-tuning methods.

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
