**Table 5:** Quantitative evaluation of MBDL models pruned by SPADE with different fine-tuning strategies in CS-MRI at the sampling rate of 10%.

| Network | Pruning Ratio | PSNR (dB) | | | SSIM (%) | | | Time (ms) | Speed Up | # Params |
|---|---|---|---|---|---|---|---|---|---|---|
| | | *Supervised* | *School* | *Self-supervised* | *Supervised* | *School* | *Self-supervised* | | | |
| DEQ | 0 % | | 35.98 | | | 96.2 | | 4954.30 | ×1.00 | 999,429 |
| | 10 % | 35.90 | 35.86 | 32.20 | 96.1 | 96.1 | 93.7 | 4812.57 | ×1.03 | 878,644 |
| | 20 % | 35.90 | 35.66 | 31.97 | 96.1 | 96.0 | 93.6 | 4708.00 | ×1.06 | 793,159 |
| | 35 % | 35.81 | 35.35 | 29.64 | 96.1 | 95.9 | 91.1 | 4233.70 | ×1.17 | 635,311 |
| | 65 % | 35.58 | 34.38 | 27.83 | 96.0 | 95.3 | 88.3 | 3274.58 | ×1.51 | 353,328 |
| VarNet | 0 % | | 34.49 | | | 95.5 | | 161.6 | ×1.00 | 19,634,712 |
| | 10 % | 34.48 | 33.85 | 33.16 | 95.4 | 95.2 | 94.5 | 161.4 | ×1.00 | 17,567,216 |
| | 20 % | 34.41 | 33.93 | 33.01 | 95.4 | 95.2 | 94.3 | 151.8 | ×1.06 | 15,772,920 |
| | 35 % | 34.31 | 33.65 | 32.00 | 95.3 | 95.0 | 93.6 | 144.7 | ×1.12 | 12,477,088 |
| | 65 % | 33.67 | 33.00 | 29.81 | 94.9 | 94.2 | 91.3 | 122.6 | ×1.32 | 6,988,752 |
| E2E-Var | 0 % | | 37.96 | | | 97.2 | | 210.7 | ×1.00 | 20,119,610 |
| | 10 % | 37.95 | 37.53 | 35.68 | 97.2 | 97.0 | 95.5 | 209.3 | ×1.01 | 18,052,114 |
| | 20 % | 37.88 | 37.54 | 35.34 | 97.2 | 97.0 | 95.4 | 201.1 | ×1.05 | 16,257,818 |
| | 35 % | 37.70 | 37.27 | 34.65 | 97.1 | 96.9 | 95.0 | 192.8 | ×1.09 | 12,961,986 |
| | 65 % | 36.83 | 36.21 | 32.64 | 96.7 | 96.3 | 93.7 | 170.4 | ×1.24 | 7,473,650 |

**Table 6:** Quantitative evaluation of USRNet pruned by SPADE with *school* fine-tuning strategy in image super-resolution at the scale of ×4. The "*Degrad. %*" columns denote degradation percentage of PSNR values of pruned models compared to that of pruned models.

| Scale | Pruning Ratio | PSNR (dB) | | SSIM (%) | | Time (ms) | Speed Up | # Params |
|---|---|---|---|---|---|---|---|---|
| | | *School* | *Degrad. %* | *School* | *Degrad. %* | | | |
| | 0 % | 25.90 | 100.0 % | 71.7 | 100.0 % | 272.2 | ×1.00 | 17,016,016 |
| ×4 | 10 % | 25.75 | 99.4 % | 71.0 | 99.1 % | 262.8 | ×1.04 | 15,314,620 |
| | 20 % | 25.75 | 99.4 % | 71.0 | 99.0 % | 248.9 | ×1.09 | 13,730,964 |
| | 35 % | 25.72 | 99.3 % | 71.0 | 99.0 % | 241.8 | ×1.13 | 10,837,246 |
| | 65 % | 25.60 | 98.9 % | 70.5 | 98.4 % | 150.3 | ×1.81 | 6,094,762 |

## A  APPENDIX

In this appendix, we present some results that were omitted from the main paper. Table 5 presents quantitative evaluations of MBDL models pruned by SPADE in MRI at a sampling rate of 10.0%. This table shows that the *supervised* fine-tuning strategy is highly effective, resulting only up to 0.4dB PSNR degradation to achieve approximately 65% fewer parameters and up to ×1.51 speed up. Table 5 also demonstrates that *school* fine-tuning method can gain competitive performance compared to *supervised* fine-tuning method. Figure 6, Figure 7, and Figure 8 present visual results of pruned models of DEQ, VarNet, and E2E-VarNet in MRI at the sampling rate of 16.7% over different pruning ratios and diverse fine-tuning methods, respectively. These figure show that *supervised* fine-tuning method can achieve promising results over different pruning ratios. These figures also demonstrate that the *school* fine-tuning method can gain competitive performance compared to the *supervised* fine-tuning approach.

Table 6 shows quantitative evaluation of USRNet pruned by SPADE with *school* fine-tuning strategy in image super-resolution at the scale of ×4. This figure highlights that the *school* fine-tuning strategy can achieve ×1.81 speed up while maintaining less than 1.1% degradation in PSNR values.

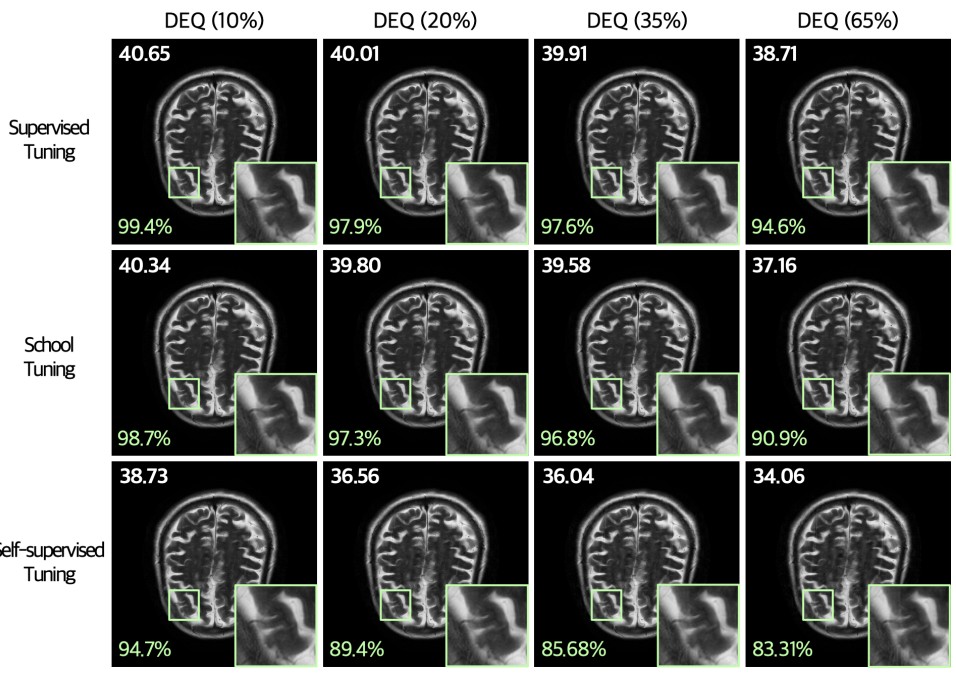

**Figure 6:** Visual results of pruned DEQ models at sampling rate of 16.7%. The PSNR values of each image with respect to the ground truth are labeled in the upper left of the image. The PSNR degradation percentage with respect to the PSNR value of the unpruned model are labeled in the bottom left of the image.

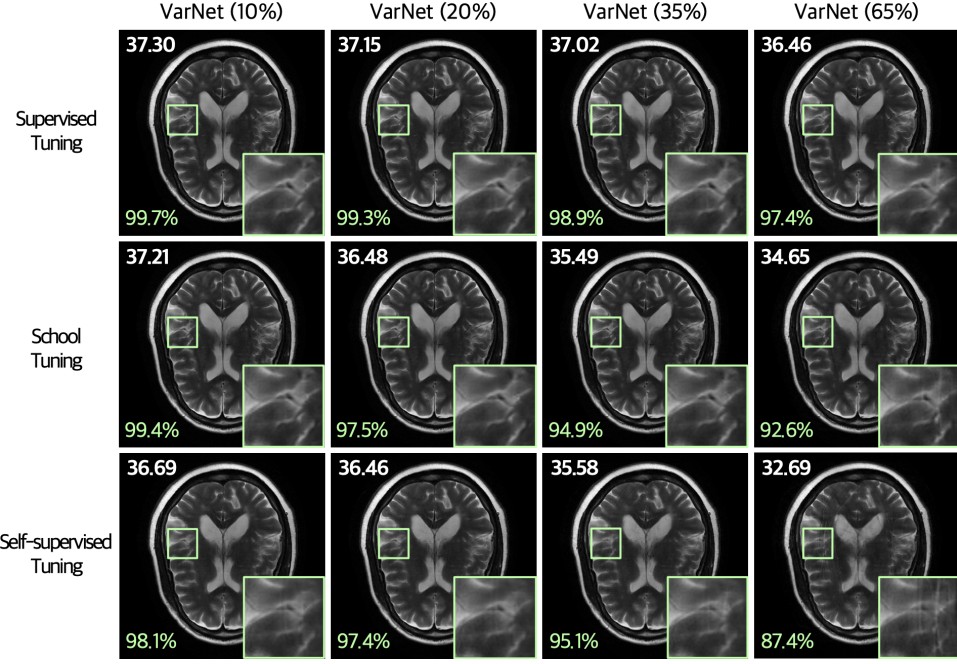

**Figure 7:** Visual results of pruned VarNet models at sampling rate of 16.7%. The PSNR values of each image with respect to the ground truth are labeled in the upper left of the image. The PSNR degradation percentage with respect to the PSNR value of the unpruned model are labeled in the bottom left of the image.

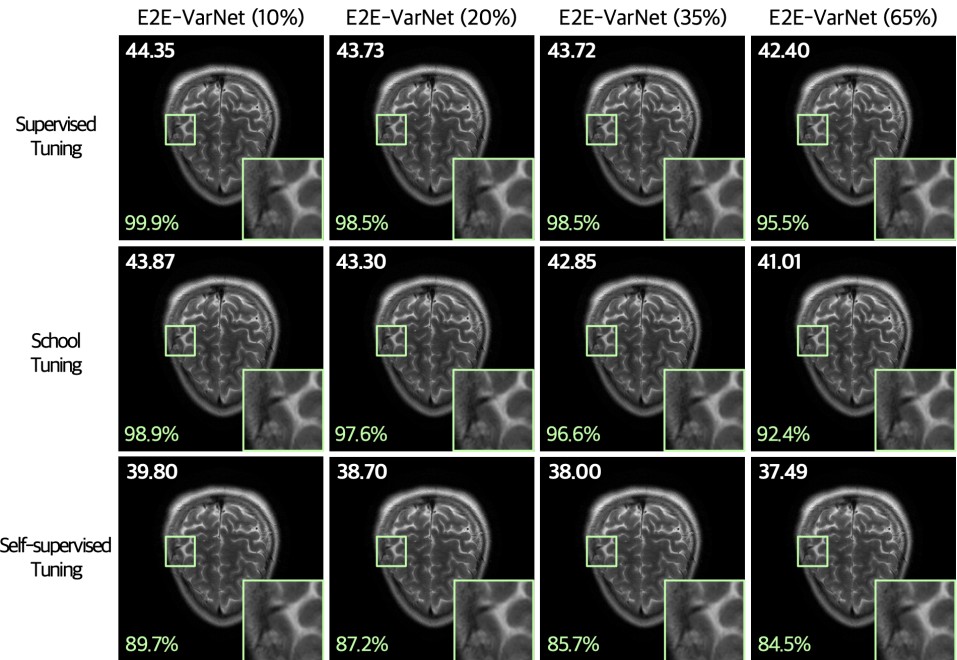

**Figure 8:** Visual results of pruned E2E-VarNet models at sampling rate of $16.7\%$. The PSNR values of each image with respect to the ground truth are labeled in the upper left of the image. The PSNR degradation percentage with respect to the PSNR value of the unpruned model are labeled in the bottom left of the image.