# OpenReview forum: "A Structured Pruning Algorithm for Model-based Deep Learning"
_ICLR.cc/2024/Conference — Submitted to ICLR 2024_

### Official Review · Reviewer_hcWa · 2023-10-24

**Soundness:** 3 good
**Presentation:** 2 fair
**Contribution:** 2 fair
**Rating:** 6
**Confidence:** 3

**Summary:**

The paper proposes a pruning approach for model based deep learning. They present structured pruning algorithm based on DepGraph and also considers three approaches to fine tune the model after the pruning to restore lost accuracy due to the pruning. They demonstrate the effectiveness of their approach by applying it to MRI and SuperResolution databases.

**Strengths:**

Motivation and experiments are well presented well. Experiments gives support to their their approach.

**Weaknesses:**

The explanation of the methods could be more clear. As I am not familiar with MBDL methods they discuss in the paper or DepGraph, I cannot really follow that how this pruning works. For example, "$f_{\theta,i}$ has dependency on $f_{\theta,j}$" is very briefly explained but it seems to be some what a key ingredient of the proposed pruning approach. And also how the layers are actually removed?   Probable not just taking a layer out of the network, as it would need some additional assumptions about compatibility of the input and output dimensions of the layer, plus not sure if network give any meaningful output after such removal. Authors could improve the presentation. For example, they could one MBDL method and describe in details that how their approach is applied to it. (e.g. Table 2 and Figure 4 can be removed or moved to a supplementary material if more space is needed)

Also, probable due to limited understanding, I fail to see that how this approach is for MBDL instead of being more general pruning method. Is there something in the pruning approach which required the model to be MBDL instead of, say, a typical classification CNN or ResNet?  I can only see that the model A is used in self-supervised fine tuning approach (Eq (10)), but its value seems quite low due to worst performance in the results. Also it could be clarified that what is their contribution or novelty with respect to DepGraph.

**Questions:**

Speed up seems quite low compared to pruning ratio. For example, 10% pruning only seems to give 0-3% speedup and E2EVar (Table1) has only 24% speedup with 65% pruning ratio. Does pruning remove lower-complexity layers, does times include possible compilation times (XLA, jit, etc), or how this can be explained?

---

> ### Author Response · Authors · 2023-11-21
> **Response to reviewer hcWa**
>
> Thanks for your feedback on our work. Please find our point-to-point responses below.
>
> > 1: The explanation of the methods could be more clear. As I am not familiar with MBDL methods they discuss in the paper or DepGraph, I cannot really follow that how this pruning works. For example, "$f_{\theta, i}$ has dependency on $f_{\theta, j}$" is very briefly explained but it seems to be some what a key ingredient of the proposed pruning approach. And also how the layers are actually removed? Probable not just taking a layer out of the network, as it would need some additional assumptions about compatibility of the input and output dimensions of the layer, plus not sure if network give any meaningful output after such removal. Authors could improve the presentation. For example, they could one MBDL method and describe in details that how their approach is applied to it. (e.g. Table 2 and Figure 4 can be removed or moved to a supplementary material if more space is needed)
>
> Prompted by your comment, the revised manuscript did a better job of explaining the pruning method. To be specific, we (1) detailed the application of SPADE to a simple network, (2) added a discussion of the inter-layer and intra-layer dependencies, and (3) provided a clear, step-by-step depiction of the pruning process. Please refer to the method section in the revised manuscript for more details.
>
> > 2: Also, probable due to limited understanding, I fail to see that how this approach is for MBDL instead of being more general pruning method. Is there something in the pruning approach which required the model to be MBDL instead of, say, a typical classification CNN or ResNet? I can only see that the model A is used in self-supervised fine tuning approach (Eq (10)), but its value seems quite low due to worst performance in the results. Also it could be clarified that what is their contribution or novelty with respect to DepGraph.
>
> While pruning is a general idea, this topic has never been explored in the context of model-based deep learning (MBDL). MBDL architectures are inherently memory intensive, where pruning can have a big impact. Our work is the first to investigate the potential of pruning in MBDL  Prompted by feedback, we revised our manuscript to better clarify our contribution. We revised the title of our paper to _“Efficient Model-based Deep Learning via Network Pruning”_ to better highlight our contribution.
>
> > 3: Speed up seems quite low compared to pruning ratio. For example, 10% pruning only seems to give 0-3% speedup and E2EVar (Table1) has only 24% speedup with 65% pruning ratio. Does pruning remove lower-complexity layers, does times include possible compilation times (XLA, jit, etc), or how this can be explained?
>
> The speed-up is not proportional to the pruning ratio because _MBDL networks consist of non-prunable data-consistency layers_ that enforce consistency between intermediate results and the raw measurements. These data-consistency layers correspond to the data-fidelity term $g(x)$ in the objective function (2). Note also recent studies have proposed to reduce the computational complexity of these data-consistency layers, such as “SGD-Net: Efficient Model-Based Deep Learning With Theoretical Guarantees”. SPADE is fully compatible with these methods.

---

> ### Comment · Reviewer_hcWa · 2023-11-21
>
> I am happy with these improvements and increased my rating. I feel that this is bit of a border case due to a bit light contribution (as I understood there is not that big methodological novelty here), but leaning slightly towards acceptance.

---

> > ### Author Response · Authors · 2023-11-22
> > **Response to reviewer hcWa**
> >
> > Thanks for the positive feedback with the updated score. If there is anything else we can clarify for you, please feel free to let us know.

---

### Official Review · Reviewer_NVrb · 2023-10-30

**Soundness:** 2 fair
**Presentation:** 3 good
**Contribution:** 2 fair
**Rating:** 5
**Confidence:** 2

**Summary:**

The authors present structured pruning algorithm for model-based deep learning (SPADE)  as a solution to reduce the computational complexity of CNNs used within Model Based Deep Learning (MBDL) framework. SPADE has two components:
1. Network Pruning: The authors adopt DepGraph to construct the dependency group for each layer in the network and use group `1-norm as the criteria to rank the importance of the filters. The non-essential weights are pruned, resulting in a reduced network structure while preserving the model's accuracy.
2. Network Fine-Tuning: Authors propose to fine-tune the pruned network to minimize performance gap. They propose three fine-tuning strategies, each with a unique benefit that depends on the presence of a pre-trained model and a high-quality ground truth.

Further, authors validate SPADE compressed sensing MRI and image super-resolution, with results showing that they can achieve substantial speed up in testing time while maintaining performance.

**Strengths:**

+ Inverse problems are theoretically interesting, and practically very relevant. Reducing inference computational complexity has help bring deep learning based solutions to the main stream for inverse problems, and potentially unlock more applications and widespread usage.

+ Authors use DepGraph (Fang et al., 2023) to perform to pruning, and show with experiments that just the pruned network has significant performance degradations. To combat this, authors propose to fine-tune the pruned network.

+ Authors propose three fine-tuning strategies for the network, which can comprehensive to cover different cases:
1. Supervised: where ground truth data is used to fine-tune.
2. School: when labels are not available, pre-trained network is used to score the test data, and in fine-tuning the discrepancy between the pruned network and original network is reduced.
3. Self-supervised: when pre-trained network and labelled data points are not available, self-supervised objective functions are used to fine-tune the pruned network.

+ Authors also perform extensive numerical experiments and evaluate SPADE on compressed sensing and super resolution, showing competitive performance while improving the computational speed.

**Weaknesses:**

+ While I appreciate the authors effort in improving the computation efficiency of deep learning methods for inverse problems, I wonder if ICLR is the correct venue for this paper. I personally think that the contributions maybe more of relevance to a compressed sensing, and super resolution community, as the pure technical contribution in this paper maybe limited. At the heart, the paper applies existing pruning method, and existing fine-tuning/training methods.

+ While the paper specialize in Model Based Deep Learning, and the introduction focuses on methods like RED, I fail to see how the proposed method SPADE is relevant to such methods, or how any specialized knowledge from this domain is used. It occurs to me that this can be applied to any CNN, not just the ones used within MBDL framework. Can authors confirm?

+ Why was compressed sensing and super resolution selected as the tasks of choice? Showing more inverse problems, and including more networks can help establishing a stronger case for the paper.

**Questions:**

Please see above.

---

> ### Author Response · Authors · 2023-11-21
> **Response to reviewer NVrb**
>
> Thanks for your feedback on our work. Please find our point-to-point responses below.
>
> > 1: While I appreciate the authors effort in improving the computation efficiency of deep learning methods for inverse problems, I wonder if ICLR is the correct venue for this paper. I personally think that the contributions maybe more of relevance to a compressed sensing, and super resolution community, as the pure technical contribution in this paper maybe limited. At the heart, the paper applies existing pruning method, and existing fine-tuning/training methods.
>
> Our main goal in this paper is to improve the efficiency of MBDL networks via network pruning. This directly relates to the computational efficiency and scalability issues of DL research, aligning with the interests of the ICLR community.
>
> > 2: While the paper specialize in Model Based Deep Learning, and the introduction focuses on methods like RED, I fail to see how the proposed method SPADE is relevant to such methods, or how any specialized knowledge from this domain is used. It occurs to me that this can be applied to any CNN, not just the ones used within MBDL framework. Can authors confirm?
>
> While pruning is a general idea, this topic has never been explored in the context of model-based deep learning (MBDL). MBDL architectures are inherently memory intensive, where pruning can have a big impact. Our work is the first to investigate the potential of pruning in MBDL  Prompted by feedback, we revised our manuscript to better clarify our contribution. We revised the title of our paper to _“Efficient Model-based Deep Learning via Network Pruning”_ to better highlight our contribution.
>
> > 3: Why was compressed sensing and super resolution selected as the tasks of choice? Showing more inverse problems, and including more networks can help establishing a stronger case for the paper.
>
> SPADE is applicable to any inverse problems and any MBDL. We considered CS-MRI and image super-resolution because they are two widely studied inverse problems. We agree that it is beneficial to validate SPADE on more inverse problems and MBDL networks. We can do it in our future work.

---

### Official Review · Reviewer_vSdx · 2023-11-01

**Soundness:** 2 fair
**Presentation:** 2 fair
**Contribution:** 1 poor
**Rating:** 3
**Confidence:** 4

**Summary:**

This work proposes a pruning algorithm using DepGraph and group L1 norm as well as three fine-tuning methods. The proposed method was evaluated in multiple experiments such as MRI and super resolution.

**Strengths:**

The proposed method showed good performance in speed up for CS-MRI (see Fig. 2) and super resolution (see Table 4).

**Weaknesses:**

It is questionable that the proposed method is indeed for MBDL only. It could be seen as a mere combination of DepGraph and group sparsity based pruning.
Three fine-tuning methods do not look novel - they are simply three usual losses that were used for inverse problems.
Experiments may not be as comprehensive as it should be: it is unclear if the sampling pattern in CS-MRI is fixed for all / super resolution was not evaluated on diverse test datasets, which were frequently used for super resolution literature.

**Questions:**

Q1. it is unclear if the proposed pruning method is novel over other prior pruning works using group sparsity. See the following works:
- W Wen et al., Learning Structured Sparsity in Deep Neural Networks, NeurIPS 2016
- J Yoon & S J Hwang, Combined Group and Exclusive Sparsity for Deep Neural Networks, ICML 2017
- Y Li et al., Group Sparsity: The Hinge Between Filter Pruning and Decomposition for Network Compression, CVPR 2020
- A Kumar et al., Pruning filters with L1-norm and capped L1-norm for CNN compression, Appl Intell 51, 2021
- K Mitsuno & T Kurita, Filter Pruning using Hierarchical Group Sparse Regularization for Deep Convolutional Neural Networks, ICPR 2021
- Z Huang et al., Rethinking the Pruning Criteria for Convolutional Neural Network, NeurIPS 2021
It is hard not to see this manuscript as a work for network pruning using group sparsity, so properly discussing and comparing the above works seems essential. I can not agree with the claim "We propose the first network pruning algorithm specifically designed for MBDL models" - to me, the proposed method can be used for any task considering DepGraph and group sparsity based pruning.

Q2. It is unclear why the proposed three fine-tuning methods are novel : they are simply three popular losses that were used for full networks and there is no special treatment for pruned networks. Please clarify.

Q3. It may be true that "its potential has remained unexplored in the realm of imaging inverse problems," but I am not sure if exploring this potential is important considering that there are a number of drawbacks. For example, it is unclear if this work simulated diverse sampling patterns for CS-MRI, which is much more practical than using a single pattern. If different sampling patterns require a new pruning, then it may not be as convenient as using a full network. Moreover, it is unclear if 51-81% speed up is beneficial by sacrificing the performance by 0.77dB in CS-MRI, which may be critical for missing clinical information. More convincing arguments for the necessity on network pruning for inverse problems along with practical experiments should follow. For super resolution, 0.68dB is a huge gap and most prior works evaluated their methods in more diverse datasets.

---

> ### Author Response · Authors · 2023-11-21
> **Response to reviewer vSdx**
>
> Thanks for your feedback on our work. Please find our point-to-point responses below.
>
> > 1: It is questionable that the proposed method is indeed for MBDL only. It could be seen as a mere combination of DepGraph and group sparsity based pruning. / Q1. it is unclear if the proposed pruning method is novel over other prior pruning works using group sparsity. See the following works ….
>
> While pruning is a general idea, this topic has never been explored in the context of model-based deep learning (MBDL). MBDL architectures are inherently memory intensive, where pruning can have a big impact. Our work is the first to investigate the potential of pruning in MBDL  Prompted by feedback, we revised our manuscript to better clarify our contribution. We revised the title of our paper to _“Efficient Model-based Deep Learning via Network Pruning”_ to better highlight our contribution.
>
> > 2: Experiments may not be as comprehensive as it should be: it is unclear if the sampling pattern in CS-MRI is fixed for all / super resolution was not evaluated on diverse test datasets, which were frequently used for super resolution literature.
>
> We used a uniform sampling pattern in CS-MRI. We conducted experiments on image super-resolution on a smaller testing set, simulating situations where only pre-trained models are accessible online and specific applications have limited data. The revised manuscript has been updated to clarify them.
>
> > 3: Three fine-tuning methods do not look novel - they are simply three usual losses that were used for inverse problems. / It is unclear why the proposed three fine-tuning methods are novel : they are simply three popular losses that were used for full networks and there is no special treatment for pruned networks. Please clarify.
>
> We would like to highlight that our paper proposes _a novel application of structured pruning to MBDL networks_, an application _not explored in any previous studies_. While similar loss functions might have been proposed in other papers, those fine-tuning methods have never been investigated in the context of efficient MBDL networks via network pruning.
>
> | Fine-tuning methods          | Need ground-truth? | Need trained original model?|
> |:--------------------:|:------------:|:------------:|
> | Supervised   | &#9745;          | &#9744;          |
> | School    | &#9744;          | &#9745;          |
> | Self-supervised        | &#9744;          | &#9744;          |
>
>
> > 4: It may be true that "its potential has remained unexplored in the realm of imaging inverse problems," but I am not sure if exploring this potential is important considering that there are a number of drawbacks. For example, it is unclear if this work simulated diverse sampling patterns for CS-MRI, which is much more practical than using a single pattern. If different sampling patterns require a new pruning, then it may not be as convenient as using a full network. Moreover, it is unclear if 51-81% speed up is beneficial by sacrificing the performance by 0.77dB in CS-MRI, which may be critical for missing clinical information. More convincing arguments for the necessity on network pruning for inverse problems along with practical experiments should follow. For super resolution, 0.68dB is a huge gap and most prior works evaluated their methods in more diverse datasets.
>
> We would like to highlight that it is a trade-off in pruning between performance and network complexity. _One can tune this trade-off based on the specific application_. For example, in applications that are sensitive to performance drop, one can set the pruning ratio in CS-MRI to 20% such that the performance drop is mostly negligible. We will improve this trade-off in our future work. It is also worth mentioning that, in Figures 5 and 6, _the visual difference is unnoticeable_ between the images recovered by the unpruned model and SPADE for CS-MRI and super-resolution, respectively.

---

### Official Review · Reviewer_ifBm · 2023-11-04

**Soundness:** 3 good
**Presentation:** 3 good
**Contribution:** 2 fair
**Rating:** 5
**Confidence:** 4

**Summary:**

This paper proposes a structured pruning method for model-based deep learning in inverse problems. The proposed method, SPADE, reduces the computational complexity of model-based networks at test-time by pruning its non-essential weights. In addition, three different fine-tuning methods are introduced for the pruned networks to reduce performance loss. SPADE is evaluated on compressed sensing MRI and image super-resolution, and is shown to speed up inference with minimal performance degradation.

**Strengths:**

Strengths:
- The application of structured pruning to model-based networks in inverse problems is new and promising, especially due to high computational costs in large-scale imaging settings.
- The proposed method results in faster inference speed and is applied to multiple frameworks, namely deep equilibrium models (DEQ) and deep unrolling (DU).

**Weaknesses:**

Weaknesses:
- The contributions of the paper are mostly comprised of a combination of existing techniques such as the pruning algorithm and the fine-tuning techniques.
- The method is not compared with other methods for improving inference speed, such as [1] or [2] mentioned in the paper. The lack of this comparison makes it difficult to quantify the significance of the results. As an example, there is a 0.77 dB PSNR drop with a 51% speed up at test-time (Table 1) for compressed sensing MRI which seems to be a large performance reduction, and it is unclear how this compares to existing methods.

[1] J. Liu, Y. Sun, W. Gan, X. Xu, B. Wohlberg, and U. S. Kamilov. SGD-Net: Efficient Model-Based
Deep Learning With Theoretical Guarantees. IEEE Trans. Computational Imag., 7:598–610,
2021.

[2] J. Tang and M. Davies. A fast stochastic plug-and-play ADMM for imaging inverse problems. arXiv
preprint arXiv:2006.11630, 2020.

**Questions:**

- How much does the training time increase for SPADE, compared with the baseline unpruned model-based network?
- Is it possible to combine fine-tuning losses, rather than view them as independent techniques, and could that help preserve performance?
- How does the memory complexity change at test-time? Memory complexity is also a quite important consideration for which discussion has not been included.

Suggestions:

- The introduction, and the "DL and MBDL." subsection in the background are repetitive. For instance, the equation for PnP/RED does not seem to contribute to the story of the paper. The background can be shortened to include more experiments in the main paper, such as the visual results (Figure 6-8) in the supplemental, which are crucial for compressed sensing MRI.
- Typographical errors should be fixed via proofreading.

---

> ### Author Response · Authors · 2023-11-21
> **Response to reviewer ifBm (1/2)**
>
> Thank you for your feedback on our work. Please find our point-to-point responses below.
>
> > 1: The contributions of the paper are mostly comprised of a combination of existing techniques such as the pruning algorithm and the fine-tuning techniques.
>
> The topic of pruning has never been explored in the context of model-based deep learning (MBDL). MBDL architectures are inherently memory intensive, where pruning can have an impact. Our work is really the first to investigate the potential of pruning in MBDL  Prompted by your remark, we revised our manuscript to better clarify our contribution. We revised the title of our paper to _“Efficient Model-based Deep Learning via Network Pruning”_ to better highlight our contribution.
>
> > 2: The method is not compared with other methods for improving inference speed, such as [1] or [2] mentioned in the paper. The lack of this comparison makes it difficult to quantify the significance of the results. As an example, there is a 0.77 dB PSNR drop with a 51% speed up at test-time (Table 1) for compressed sensing MRI which seems to be a large performance reduction, and it is unclear how this compares to existing methods.
>
> We would like to highlight that our approach is _fully compatible_ with the algorithms you mentioned. These methods focus on operators related to the data fidelity term in the objective function, while our approach focuses on the neural network prior. One can easily integrate these algorithms into SPADE to further improve efficiency, but it is out of the scope of our study.
>
> > 3: How much does the training time increase for SPADE, compared with the baseline unpruned model-based network?
>
> Prompted by your comment, we compared the time of training the original VarNet and fine-tuning its pruned variant at a 65% pruning ratio. As shown in the table below, fine-tuning times in SPADE are much shorter than that for training the original model.
>
> **SPADE Fine-tuning Time (h)**
> | Network | Pruning Ratio |      Supervised       |        School        |   Self-supervised    |
> |:-------:|:-------------:|:---------------------:|:--------------------:|:--------------------:|
> | VarNet  |      0%       |     446.26 (100%)     |                      |                      |
> |         |      65%      | 51.48 (11.5%)  | 5.39 (1.21%)   | 10.38 (2.3%)  |
>
> > 4: Is it possible to combine fine-tuning losses, rather than view them as independent techniques, and could that help preserve performance?
>
> Thanks for your suggestions. We indeed have _experimented with combined fine-tuning losses before but did not find any improvement_. Our hypothesis is that the hierarchy of supervision quality—supervised, school, and self-supervised fine-tuning—may lead to one loss function dominating the supervision when combined. As an example, we formulated below a combined loss function with the addition of school loss and self-supervised loss.
>
> $l_{combined}(\hat{\theta}) = l_{sc}(\hat{\theta}) + l_{ss}(\hat{\theta})$
>
> We conducted experiments on the VarNet model with a 65% pruning ratio. The fine-tuning results
> below show that using combined loss does not have an advantage over solely employing school fine-tuning losses.
>
> **Quantitative evaluation of three different losses: combined, school, and self-supervised in PSNR**
> | Network | Pruning Ratio | School + Self-supervised | School | Self-supervised |
> |:---------:|:---------------:|:--------------------------:|:--------:|:-----------------:|
> | VarNet  | 0%            | 39.25                    | -      | -               |
> | VarNet  | 65%           | **36.88**                   | 37.36  | 34.17           |
>
> **Quantitative evaluation of three different losses: combined, school, and self-supervised in SSIM(%)**
> | Network | Pruning Ratio | School + Self-supervised | School | Self-supervised |
> |:---------:|:---------------:|:--------------------------:|:--------:|:-----------------:|
> | VarNet  | 0%            | 97.7                     | -      | -               |
> | VarNet  | 65%           | **96.6**                     | 96.7   | 95.3            |

---

> ### Author Response · Authors · 2023-11-21
> **Response to reviewer ifBm (2/2)**
>
> > 5: How does the memory complexity change at test-time? Memory complexity is also a quite important consideration for which discussion has not been included.
>
> Prompted by your comment, we evaluated the test-time memory complexity across different pruning ratios for DEQ, VarNet, and E2EVar networks. The results as shown below demonstrate a clear reduction in GPU memory usage as the pruning ratio increases.
>
> | Network | Pruning Ratio | GPU Memory (MB) |
> |:---------:|:---------------:|:-----------------:|
> | DEQ     | 0%            | 3.82 (100.0%)   |
> |         | 10%           | 3.37 (88.22%)   |
> |         | 20%           | 3.04 (79.58%)   |
> |         | 35%           | 2.44 (63.87%)   |
> |         | 65%           | 1.37 (35.86%)   |
> | VarNet  | 0%            | 76.37 (100.0%)  |
> |         | 10%           | 68.06 (89.12%)  |
> |         | 20%           | 61.16 (80.08%)  |
> |         | 35%           | 47.96 (62.80%)  |
> |         | 65%           | 26.77 (35.05%)  |
> | E2EVar  | 0%            | 77.59 (100.0%)  |
> |         | 10%           | 70.27 (90.57%)  |
> |         | 20%           | 63.77 (82.19%)  |
> |         | 35%           | 49.72 (64.08%)  |
> |         | 65%           | 28.83 (37.16%)  |
>
> > 6: The introduction, and the "DL and MBDL." subsection in the background are repetitive. For instance, the equation for PnP/RED does not seem to contribute to the story of the paper. The background can be shortened to include more experiments in the main paper, such as the visual results (Figure 6-8) in the supplemental, which are crucial for compressed sensing MRI.
>
> Thanks for your suggestion. The equation of PnP/RED holds significance for introducing key concepts and the updates in iterations of DU and DEQ. Omitting the PnP/RED equation necessitates an alternative equation for the update iterations of DU/DEQ. We attempted to incorporate the MRI figure into the main paper as per your suggestions. However,  the oversized nature of the MRI figures prevents their inclusion.
>
> > 7: Typographical errors should be fixed via proofreading.
>
> The revised manuscript has done a better job of fixing the typographical errors.

---

### Author Response · Authors · 2023-11-21
**Response to all reviewers:**

Thank you all for providing us with valuable feedback. We provide point-to-point responses to all the comments below. We updated the manuscript and the supplement with revisions highlighted in red color. In particular, we revised the title to _“Efficient Model-based Deep Learning via Network Pruning”_ to better highlight our contribution.

---

### Author Response · Authors · 2023-11-22
**Response to all reviewers**

Dear all, thank you again for reading our paper. We are nearing the end of the discussion period. Please let us know if there is anything else we can do to further improve the paper or answer any questions.

---

### Meta-Review · Area_Chair_vu4s · 2023-12-03

**Metareview:**

The paper proposes a pruning method for model based deep learning algorithms for image reconstruction problems. The method reduces the computational cost by pruning non-essential weights. The method is evaluated on accelerated MRI and image super-resolution and improves speed with a small performance degradation.

* Reviewer ifBM (5) finds the application of pruning to model-based networks promising, and acknowledges that the method results in faster inference speeds. However, the reviewer points out a lack of comparisons: pruning improves computational cost, but lowers performance; perhaps training a smaller model from the beginning is better. In the rebuttal, the authors show advantages for training time; however training times for image reconstruction methods are not a bottleneck, speed at inference is more important.

* Reviewer vSdx (3) notes that the method shows good performance, but is worried about novelty/contribution. The authors respond that the application to model based imaging is new.

* Reviewer NVrb (5) is also concerned about novelty since the paper at heart applies an existing pruning method.

* Reviewer hcWa (6) finds the motivation and experiments well presented; but also notes that the paper is applying a relatively general pruning approach to model based imaging.

The paper's contributions is limited to applying a relatively known pruning approach to model based method, and the benefits that this provides is not sufficiently well characterized. In particular, it is unclear whether pruning leads to benefits, or whether training smaller models might not give better results.

**Justification For Why Not Higher Score:**

Limited comparisons and limited contribution.

**Justification For Why Not Lower Score:**

N/A

---

### Decision · Program_Chairs · 2024-01-16

Reject